# Alkaloid Derivative (*Z*)-3β-Ethylamino-Pregn-17(20)-en Inhibits Triple-Negative Breast Cancer Metastasis and Angiogenesis by Targeting HSP90α

**DOI:** 10.3390/molecules27207132

**Published:** 2022-10-21

**Authors:** Xin-Yao Liu, Yu-Miao Wang, Xiang-Yu Zhang, Mei-Qi Jia, Hong-Quan Duan, Nan Qin, Ying Chen, Yang Yu, Xiao-Chuan Duan

**Affiliations:** 1School of Pharmacy, Tianjin Medical University, Tianjin 300070, China; 2Research Center of Basic Medical Sciences, Tianjin Medical University, Tianjin 300070, China; 3Key Laboratory of Immune Microenvironment and Disease (Ministry of Education), Tianjin Medical University, Tianjin 300070, China; 4School of Biomedical Engineering and Technology, Tianjin Medical University, Tianjin 300070, China

**Keywords:** compound **1**, triple-negative breast cancer, antimetastasis, antiangiogenesis, HSP90α, HIF-1α/VEGF/VEGFR2

## Abstract

Metastasis is an important cause of cancer-related death. Previous studies in our laboratory found that pregnane alkaloids from *Pachysandra terminalis* had antimetastatic activity against breast cancer cells. In the current study, we demonstrated that treatment with one of the alkaloid derivatives, (*Z*)-3β-ethylamino-pregn-17(20)-en (**1**), led to the downregulation of the HIF-1α/VEGF/VEGFR2 pathway, suppressed the phosphorylation of downstream molecules Akt, mTOR, FAK, and inhibited breast cancer metastasis and angiogenesis both in vitro and in vivo. Furthermore, the antimetastasis and antiangiogenesis effects of **1** treatment (40 mg/kg) were more effective than that of Sorafenib (50 mg/kg). Surface plasmon resonance (SPR) analysis was performed and the result suggested that HSP90α was a direct target of **1**. Taken together, our results suggested that compound **1** might represent a candidate antitumor agent for metastatic breast cancer.

## 1. Introduction

Breast cancer is the most common cancer type for women [1]. According to a recently published report, about 70,700 people died from breast cancer in 2015, mostly from metastatic breast cancer [2]. About 6–10% of breast cancer patients have metastatic breast cancer at their initial diagnosis, while 20–30% of patients with lower-stage breast cancer will develop metastases. There has been a significant effort to investigate and develop therapies against cancer metastasis in the last decades. However, metastasis remains a largely untreatable disease.

Hypoxia-inducible factor-1α (HIF-1α) is a potential target for suppressing tumor metastasis [3]. Recent studies revealed that HIF-1 is overexpressed in several solid tumors, including breast cancer [4,5]. HIF-1α is associated with several crucial aspects of cancer progression, including migration, invasion, angiogenesis and chemoresistance [6,7]. HIF-1α transcriptionally regulates a battery of genes that are pivotal for tumor angiogenesis and metastasis, including the pro-angiogenic vascular endothelial growth factors (VEGFs) [7]. The VEGFs are the most important stimulators of tumor angiogenesis [8]. Therefore, inhibiting HIF-1α is considered as an effective strategy to restrain tumor growth [3].

Heat Shock Proteins (HSPs) constitute a group of proteins that play a crucial role in protein folding. High expression of HSPs was reported in many cancers, such as breast, lung, colorectal, prostate, ovarian and gastric cancer. A number of investigations suggested that HSPs were promising hallmarks in cancers. These proteins helped in the proliferation, invasion and metastasis of tumor cells [9].

Several of HSPs, such as HSP40, HSP60, HSP70 and HSP90, promoted tumor cell proliferation and metastasis. Yang et al. investigated the functional mechanism of HSP40 in cancer. Their research demonstrated that DnaJA1 (Hsp40) is transcribed by E2F transcription factor 1 and promotes cell cycle progression by inhibiting ubiquitin degradation of cell division cycle protein 45 in colorectal cancer [10]. HSP60 promote cancer cell growth through regulating mitochondrial biogenesis. In glioblastoma, inhibition of HSP60 increased the level of reactive oxygen species in mitochondria, subsequently leading to the AMPK activation, which in turn, suppressed phosphorylation of S6K and 4EBP1 and inhibited cancer cell growth [11]. HSP70 was involved in cancer cell epithelial to mesenchymal transition progression and invasion. It was reported that increased levels of HSP70 contribute to breast cancer metastasis by upregulating N-cadherin, MMP2, SNAIL and vimentin [12]. In addition, Garg et al. reported that the germ cell-specific protein HSP70, which is expressed in 86% of cervical cancer specimens, was related to cancer cell growth, colony formation, migration and invasion [13].

Recent investigations have reported that HIF-1α was a client protein of heat shock protein 90 (HSP90) [14]. HSP90 is a ubiquitous molecular chaperone that ensures proper folding, assembly and maturation of its client proteins, which is crucial for their functions and activities. Through interacting with them, HSP90 regulates several key cellular processes including cell proliferation, cell cycle, signaling transduction as well as tumor progression [15,16]. It has been reported that HSP90 is frequently upregulated in various types of cancer. Inactivation of HSP90 could result in the degradation of its client proteins and apoptosis of tumor cells. Hence, HSP90 is emerging as a promising drug target for inhibiting tumor progression [16,17,18,19].

*Pachysandra terminalis* Sieb. et Zucc. (Buxaceae) is a medicinal plant mainly distributed in the southwestern region of China and Japan. Previous studies on the chemical composition of *Pachysandra terminalis,* performed by our group, revealed that it contained a pregnane alkaloid that inhibited breast cancer cell chemotaxis [20]. Using the pregnane alkaloid as a lead compound, we synthesized a series of novel alkaloid derivatives and evaluated their effects on breast cancer cell migration [21]. One of the compounds, (*Z*)-3β-ethylamino-pregn-17(20)-en (**1**), had an IC50 as low as 0.17 μM and was selected for further evaluation as a potential agent against human metastatic breast cancer.

In this study, we examined the effects of **1** on the metastasis and angiogenesis of breast cancer and its molecular mechanism in vitro and in vivo. Our results showed that compound **1** could inhibit several processes involved in breast cancer metastasis, including adhesion, migration, invasion, and suppression angiogenesis. Furthermore, we demonstrated that **1** affected these angiogenesis and metastatic processes by suppressing the HIF-1α/VEGF/VEGFR2 axis and its downstream signaling molecules. We further demonstrated that HSP90α is a direct target of **1**.

## 2. Results

### 2.1. Cytotoxicity of Compound ***1*** against MDA-MB-231 Cells and Huvecs

Non-toxic doses of **1** (chemical structure shown in Figure 1) were determined for the MDA-MB-231 cells and HUVECs using the MTT assay. Compound **1** had no obvious cytotoxic effects on MDA-MB-231 cells or HUVECs when administered at concentrations less than 5 μM (Appendix A), respectively. The remaining experiments presented in this paper were performed using non-cytotoxic concentrations of **1** to exclude the influence of cell viability, except otherwise stated.

### 2.2. Compound ***1*** Suppressed Cell Adhesion

The adhesion of cancer cells to the extracellular matrix is the initial step of cell migration and invasion, which are associated with both metastasis and angiogenesis. We found that **1** markedly inhibited the adhesion of both MDA-MB-231 cells or HUVECs in a dose-dependent manner (Figure 2A,B).

### 2.3. Compound ***1*** Suppressed Cell Migration and Invasion

We performed a wound-healing assay to assess the impact of **1** on cell migration. Treatment with increasing concentrations of **1** caused concomitant suppression of MDA-MB-231 cells (Figure 2C) and HUVECs (Figure 2D) migration.

Invasion through the basement membrane is another essential step for cancer metastasis. We used the Matrigel-coated transwell assay to evaluate the effect of **1** on cell invasion. Cells that invaded through the Matrigel in 24 h were stained, photographed and counted. Compared to the control, cells treated with **1** had a markedly reduced invasive potential in both cell types. This effect was dose-dependent (Figure 2E,F).

### 2.4. Compound ***1*** Suppressed HUVECs Tube Formation

Angiogenesis is the formation of new blood vessels that provide the oxygen and nutrients necessary for tumor cell growth and metastasis. HUVECs can form three-dimensional tubular structures in Matrigel, which could be enhanced by VEGF stimulation. Therefore, it is an appropriate model for evaluating the effect of a compound on angiogenesis. Using this model, we examined the ability of **1** to abrogate in vitro tube formation. This assay demonstrated significant dose-dependent inhibition of the HUVECs tube formation by **1** (Figure 3).

### 2.5. Compound ***1*** Downregulated Intracellular and Extracellular VEGF Levels

VEGF family members are the most important and best-characterized modulators of angiogenesis [8]. This family consists of five members, VEGF-A–D and placental growth factor. Among these members, VEGF-A is the major one that stimulates angiogenesis and is commonly referred to as VEGF [22]. VEGF can be secreted by various cell types, including cancer cells. Several studies have suggested that VEGF levels are correlated with the metastatic growth of human cancers, including breast cancer [23,24]. Therefore, we examined the effects of **1** on VEGF expression and secretion in MDA-MB-231 cells. As shown in the figures, compound **1** suppressed both the intracellular expression (Figure 4A) and extracellular secretion (Figure 4B) of VEGF under both normoxia and hypoxia.

### 2.6. Compound ***1*** Inhibited Hypoxia-Induced HIF-1α Protein Expression

HIF-1α is the most important transcription factor regulating VEGF expression and an essential stimulator of tumor development and metastasis [3]. Thus, we assessed the impact of **1** on HIF-1α protein expression using MDA-MB-231 breast cancer cells. As shown in Figure 5A, under hypoxic and serum starvation conditions, compound **1** downregulated HIF-1α levels in a dose-dependent manner.

### 2.7. The Inhibitory Effect of ***1*** Was Abolished by an HIF1-α Activator

To confirm the role of **1** as a HIF-1α inhibitor, we examined the effects of the HIF-1α activator dimethyloxallyl glycine (DMOG) [25] on compound **1**-mediated inhibition of MDA-MB-231 cell migration using the wound-healing assay. The cells were pre-treated with or without DMOG (100 μM) and then exposed to 5 μM of **1.** Untreated cells served as a negative control. Treatment with DMOG rescued MDA-MB-231 cell migration from the suppressive effects of **1** (Figure 5B). These data suggested that **1** inhibited cell migration mainly through the suppression of HIF-1α activity.

### 2.8. Compound ***1*** Inhibited the Activation of VEGFR2 and Downstream Signaling Pathways In Vitro

The function of VEGF is mediated by its binding to its cognate receptors (vascular endothelial growth factor receptors, VEGFRs). The VEGFRs represent a receptor tyrosine kinase family comprised of three members, VEGFR1–3. Among these, VEGFR2 is the major one contributing to tumor angiogenesis [26,27]. The binding of VEGF to VEGFR2 results in the phosphorylation and activation of VEGFR2, triggering a signaling cascade that activates several signaling pathways associated with cell proliferation and migration [28]. Therefore, we evaluated the effects of **1** on VEGFR2 and its downstream signaling pathways.

In MDA-MB-231 cells, treatment with **1** significantly inhibited the activation of VEGFR2, resulting in the suppression of its downstream signaling molecules including Akt, mTOR and FAK phosphorylation. These effects were dose-dependent. (Figure 6). Compound **1** also suppressed VEGFR2 activation in HUVECs with the stimulation of VEGF (20 ng/mL), resulting in concomitant downregulation of Akt, mTOR and FAK phosphorylation (Figure 7).

### 2.9. Compound ***1*** Inhibited the Tumor Growth, Metastasis and Angiogenesis in 4T1 Mammary Carcinoma Models

The orthotopic 4T1 mammary carcinoma models were used to investigate the antitumor, antimetastasis and antiangiogenesis effects of **1**. As shown in Figure 8A and 8B, tumor growth was significantly inhibited by the treatment of 50 mg/kg Sorafenib and 40 mg/kg compound **1** compared with the physiological saline group. Moreover, treatment of 40 mg/kg compound **1** significantly inhibited the tumor growth compared with the 50 mg/kg Sorafenib group. The tumor weight at the end of the experiment (Figure 8C) also demonstrated that 40 mg/kg of **1** resulted in a significantly lower tumor weight than that of Sorafenib. Bioluminescence analysis (Figure 9A) of each group showed similar results with the tumor growth curve. Both the 50 mg/kg Sorafenib and 40 mg/kg compound **1** treatment could remarkably suppress orthotopic tumor growth at day 29. In addition, as shown in Figure 8D, no significant changes in body weight were recorded for all the treatment groups throughout the tumor inhibition experiment in 4T1 tumor-bearing mice.

Bioluminescence analysis (Figure 9A) of each group showed that both Sorafenib at 50 mg/kg and compound **1** at 40 mg/kg remarkably suppressed lung metastasis and the BLI signal among lungs could hardly be observed at day 29. Then, lung metastases were further analyzed using white light imaging of lung nodules (Figure 9B). The data revealed that the number of lung metastases (Figure 9E) was reduced significantly by 50 mg/kg Sorafenib and 40 mg/kg compound **1**, compared with the saline group. Notably, there were fewer lung nodules that existed in mice treated with 40 mg/kg compound **1** than those receiving 50 mg/kg Sorafenib. These results suggested that **1** inhibited tumor growth and lung metastasis in 4T1 tumor-bearing mice in a dose-dependent manner. In addition, the H&E staining of lung tissues showed that the metastatic tumor cells in lung tissues, which were closely arranged, with large and deep nuclear staining could hardly be observed in Sorafenib (50 mg/kg) and **1** (40 mg/kg)-treated groups (Figure 9C). Furthermore, CD31 immunostaining (Figure 9D) and MVD count (Figure 9F) confirmed that treatment with Sorafenib (50 mg/ kg) and **1** (40 mg/kg) significantly reduced tumor vascularization. Compound **1** (40 mg/kg) significantly reduced the MVD of tumor tissues compared with that of Sorafenib (50 mg/kg).

Western blotting analysis were performed to confirm the antimetastatic and antiangiogenesis molecular mechanism in 4T1 mammary carcinoma models. As shown in Figure 10, **1** suppressed the expression of HIF-1α and inhibited the activation of VEGFR2, AKT/mTOR and FAK in a dose-dependent manner.

### 2.10. HSP90α Was a Direct Target of ***1***

Next, we tried to discover the molecular target of **1**. Recent investigations revealed that targeting couples of HSP90 inhibitors could downregulate HIF-1α [29,30,31]. Moreover, several above-mentioned signaling molecules, including Akt and FAK, are also client proteins of HSP90 [15]. Therefore, the molecular dynamics study was examined, firstly, to investigate the specific binding of **1** to HSP90α. As shown in Appendix A, compound **1** could directly cage into the binding pocket of HSP90α. The docking score was −8.048 in its best binding pose.

Then, we performed a cellular thermal shift assay (CETSA) to examine whether **1** could bind to HSP90. The CESTA is a novel approach to evaluate the interaction between compound and protein [32]. If a compound binds with a protein, the thermal stability of the protein will increase. Based on this, MDA-MB-231 cells were treated with either 15 μM of **1** or vehicle (0.1% DMSO) and a CETSA was performed as previously stated. SNX-2112, an HSP90 inhibitor, was used as a positive control. Results showed that **1** could increase the thermal stability of HSP90, suggesting that HSP90 is a direct target of **1** (Figure 11A). This effect is also dose-dependent (Figure 11B).

In addition, the binding affinity of **1** towards HSP90α was investigated using SPR (Figure 11C). The association (ka) and dissociation (kd) rate constants of **1** binding to HSP90α were 9930 1/Ms and 0.209 1/s, respectively. The equilibrium dissociation constant (KD) of **1** was 21.0 μM. Thus, the direct target of **1** was further demonstrated as HSP90α.

## 3. Discussion

Metastasis is a multi-step process involving several coordinated events, including the epithelial-to-mesenchymal transition (EMT), migration, adhesion, invasion and angiogenesis [33,34,35]. It is accompanied by a series of cellular and molecular changes and is regulated by multiple signaling pathways and molecules. Targeting these signaling pathways and molecules could be a potential strategy for controlling cancer progression [35].

In 1971, Folkman et al. proposed that the formation of new blood vessels in the tumor site correlated with cancer development [36]. Since then, significant attention has been focused on whether or how cancer patients could be cured by restraining angiogenesis. Through these studies, various types of angiogenesis regulators have been identified and characterized [37]. The VEGFs are the most important and well-studied factors that directly promote tumor angiogenesis by activating endothelial cell proliferation and migration [28]. Moreover, several studies have shown the role of VEGF in promoting cancer cell migration and invasion [38,39,40]. Furthermore, several recent investigations suggested that autocrine VEGF can promote the survival and progression of cancer independent of its role in angiogenesis [41,42]. These results have been confirmed by our work.

A hypoxic microenvironment is a hallmark of cancer development due to the rapid growth of cancer cells and lack of blood supply [3]. HIF-1α is a key hypoxia regulator and pro-angiogenic factor that accumulates in cancer cells under hypoxic conditions and is involved in multiple events crucial for cancer metastasis, including cancer cell migration, invasion and angiogenesis [6,7]. Under hypoxia, HIF-1α transcriptionally regulates a battery of genes that are pivotal for tumor angiogenesis and metastasis, including the VEGFs [7,43]. In particular, HIF-1α is highly expressed in triple-negative breast cancer, which is hard to treat due to the lack of a therapeutic target. Therefore, HIF-1α may be a good target for anticancer therapy. Accordingly, screening for small molecules that inhibit HIF-1α and its downstream signaling has been a trend in anticancer drug development [43]. In this study, we demonstrated that the pregnane alkaloid **1** could downregulate HIF-1α and block the VEGF/VEGFR2 axis and its downstream signaling pathways in triple-negative breast cancer cells. This effect was abolished when cells were exposed to a HIF-1α activator, suggesting that the antimetastatic activity of **1** is mediated through HIF-1α.

Identification of the molecular target is essential for elucidating the working mechanism of a compound. HIF-1α is downregulated under several circumstances, and among these, the most reported approach is through targeting HSP90 [29,31,44], which is also a hot target for antitumor agents. Our work demonstrated that compound **1** bonds directly with HSP90α by CETSA and SPR methods.

HSP90 has long been considered a promising target for antitumor agents due to its indispensable role in the regulation and stabilization of multiple cancer-related proteins and pathways. HSP90 simultaneously modulated cancer cell proliferation, invasion, metastasis, angiogenesis and resistance to apoptosis [33]. HSP90 expressed in cancer cells was shown to form 100-fold tighter multi-chaperone complexes than in non-tumor cells [45,46]. Therefore, HSP90 in cancer cells are more sensitive to inhibitor treatments. Furthermore, for the reason that HSP90 affects multiple targets and pathways, HSP90 inhibitors are less likely to induce resistance [47]. In the present study, molecular docking simulation, CETSA and SPR analysis demonstrated that compound **1** binds tightly to the HSP90 in MDA-MB-231 cells. These results suggested that compound **1** might be used as monotherapy or combined therapy to avoid the development of drug resistance in clinics.

To date, more than 20 HSP90 inhibitors have entered clinical trials for the treatment of various types of cancer but none of them were approved for clinical use due to side effects [19,48]. HSP90 inhibition by natural, semi-synthetic and synthetic compounds has yielded promising results in pre-clinical studies and clinical trials for multiple tumor types. The natural inhibitors of HSP90 include Ansamycins, Geraniin, Gambogic acid, Panaxynol, Deguelin, Heteronemin, Radicicol and pochonins [18]. These natural compounds have a remarkably modulated HSP90 activity and serve as scaffolds for the development of novel synthetic or semi-synthetic inhibitors as well.

Many natural compounds with antitumor activities from natural medicines have been reported. Zhu et al. reported that Tubocapsenolide A, a steroid form *Tubocapsicum anomalum* could inhibit the proliferation of multiple types of cancer cells by targeting C-terminal cysteine residues of HSP90 [49]. A piperidine type of alkaloid, daurisoline, suppressed lung cancer tumorigenesis and proliferation by targeting HSP 90 and destabilizing β-catenin [50]. A lot of medicinal agents derived from plants have been extensively investigated to develop anticancer drugs with fewer adverse effects. Among them, steroidal alkaloids are conventional secondary metabolites that comprise an important class of natural products [51]. Except for our previous study [20], many pregnane alkaloids exhibited significant cytotoxic activities on multifarious cancer cells [52]. In marine natural products, a steroidal alkaloid cortistatin A, from the Marine Sponge *Corticium simplex*, revealed antiangiogenic activity by inhibiting the migration and tubular development of HUVECs promoted by VEGF [52]. In this study, we demonstrated for the first time that compound **1**, a pregnane alkaloid derivative, bonds directly with HSP90α, leading to the downregulation of HIF-1α and its downstream signaling pathways and inhibition of breast cancer matastasis and angiogenesis in vitro and in vivo, Furthermore, treatment with **1** (40 mg/kg) demonstrated more efficacy in antimetastasis and antiangiogenesis than Sorafenib (50 mg/kg).

## 4. Materials and Methods

### 4.1. Agents and Antibodies

Compound **1** was synthesized as in the previous report [21]. Recombinant human HSP90 alpha protein (ab80369) was acquired from Abcam (Cambridge, MA, USA). Sorafenib was purchased from Sigma–Aldrich Inc. (St. Louis, MO, USA). 3-(4,5-Dimethylthiazol-2-yl)-2,5-diphenyltetrazolium bromide (MTT) was obtained from Macklin Biochemical Co., Ltd. (Shanghai, China). VascuLife^®^ VEGF Cell Culture Medium was provided by Lifeline Cell Techology (Frederick, MD, USA). Cell culture media L-15, DMEM, penicillin, streptomycin, FBS and L-glutamine were all obtained from Thermo Fisher Scientific (Waltham, MA, USA). Matrigel Matrix was purchased from Corning, Inc. (Corning, NY, USA). Fibronectin was given by Merck Millipore (Darmstadt, Germany). RIPA lysis buffer containing protease inhibitor was acquired from Solarbio (Beijing, China). BCA protein assay kits were purchased from Invitrogen, ThermoFisher Scientific Inc. (Waltham, MA, USA). Hematoxylin and eosin staining solution was provided by Zhong Shan Golden Bridge Biotechnology Co., Ltd. (Beijing, China).

Antibodies against β-Actin (#4970), HIF-1α (#36169), HSP90 (#4877), Akt(#4691), Phospho-Akt (Ser473) (#4060), mTOR (#2983), Phospho-mTOR (Ser2448) (#5536), FAK (#71433), Phospho-FAK (Tyr397) (#8556), VEGF Receptor 2 (#9698) and Phospho-VEGF Receptor 2 (Tyr1175) (#2478), were purchased from Cell Signaling Technology, Inc. (Danvers, MA, USA).

### 4.2. Cell Culture and Treatment

The human MDA-MB-231 breast cancer cell line was from the American Type Culture Collection (Rockville, MD) and cultured in an L15 medium containing 10% fetal bovine serum (Bioind, Kibbutz Beit-Haemek, Israel). Human umbilical vein endothelial cells (HUVECs) were a kind gift from the General Hospital of Tianjin Medical University. The HUVECs were cultured in VascuLife basal medium supplemented with cytokines. Mouse mammary carcinoma (4T1) cells were obtained from the Chinese Academy of Sciences Cells Bank (Shanghai, China) and cultured in Dulbecco’s modified Eagle’s medium (DMEM) (high glucose) medium with 10% fetal bovine serum. Firefly luciferase-labeled 4T1 cells (4T1-Luc) were produced using Recombinant Lentivirus (GenePharma, Shanghai, China) according to the manufacturer’s instructions. For hypoxic cell culture, cells were grown in an L15 starvation medium containing1% FBS for 12 h before the administration of **1**. The cells were then placed into a hypoxic incubator filled with high-purity nitrogen (>99.999%).

Compound **1** was dissolved in dimethyl sulfoxide (DMSO) and, subsequently, a sterile medium for cell treatment. The final concentration of DMSO was 0.1%.

### 4.3. Cell Viability Assay

Cell viability was measured using the MTT assay. MDA-MB-231 cells and HUVECs (1 × 10^4^/well) were seeded in 96-well plates (BD Biosciences, Bedford, MA, USA). The next day, the cells were treated with various concentrations of **1**. After 48 h, MTT (0.5 mg/mL) was added to each well, and the cells were incubated for an additional 4 h. The OD value was read at 490 nm.

### 4.4. Cell Adhesion Assay

For the adhesion assay, 96-well plates were coated with fibronectin (20 μg/mL) at 4 °C overnight and further blocked with bovine serum albumin (BSA; 1%) for 1 h. Following pretreatment with 0.1, 1 and 5 μM of **1** for 24 h, cells were harvested, resuspended in serum-free medium containing various concentrations of **1** and seeded to the 96 wells plate coated with fibronectin at a density of 1 × 10^4^ cells/well. For HUVECs, cells were stimulated with VEGF (20 ng/mL), meanwhile. Following additional incubation for 1 h, nonadherent cells were removed using PBS washes. Cell adherence was fixed and stained, and the OD values of each well were measured using a microplate reader at 570 nm. Data are presented relative to the untreated control.

### 4.5. Migration and Invasion Assays

Cells were seeded in 6-well plates and allowed to reach a full monolayer. A clean wound was made by scratching across the center of the wells using sterile pipette tips. After washing with PBS, the cells were treated with **1** (0.1, 1, 5 μM) for 16 h. For HUVECs, cells were stimulated with VEGF (20 ng/mL) meanwhile. The wound before and after cell migration was photographed under a microscope, and the width of the wound was measured. The migration distance was then calculated for each well.

For the cell invasion assay, transwell inserts (6.5 mm, Costar, Manassas, VA, USA) were coated with 1 mg/mL Matrigel. Cells were pre-treated with increasing concentrations of **1** for 24 h. For HUVECs, cells were stimulated with VEGF (20 ng/mL), meanwhile. Cell suspensions (2 × 10^5^ cells treated with different concentrations of **1** in 100 μL) were plated in the upper chamber. The lower wells contained 750 μL medium supplemented with 10% serum. After 24 h, the cells that invaded through the Matrigel were fixed and stained. Three fields under 10× magnification were randomly selected for cell counting.

### 4.6. Tube Formation Assay

Each well of a 96-well plate was coated with 50 μL of Matrigel and incubated for 1 h at 37 °C. HUVECs (1 × 10^5^ in 100 μL) pretreated with **1** (0.1, 1, 5 μM) and VEGF (20 ng/mL) were seeded on the Matrigel and incubated at 37 °C for 12 h. Three non-overlapping fields were randomly chosen for each well and photographed (100× magnification). The total number of branching points formed by the HUVECs was calculated per field.

### 4.7. VEGF Enzyme-Linked Immunosorbent Assay (ELISA)

The extracellular and intracellular VEGF concentrations were determined using the human VEGF ELISA kit (RD Systems, Tustin, CA, USA), according to the manufacturer’s instructions. An enzyme marker (BioTek, Winooski, VT, USA) was used to determine the OD value at 450 nm.

### 4.8. Western Blotting Analysis

Whole-cell lysates were prepared in ice-cold RIPA buffer (Solarbio, Beijing, China) supplemented with a 1% protease inhibitor (PMSF). Cell lysates were centrifuged at 10,000× *g* for 20 min at 4 °C. The cell debris was discarded, and the supernatant was transferred to a new tube. Protein concentrations were determined using the BCA protein assay (Jiancheng, Nanjing, China) Protein samples (50 μg) were separated by 7% or 10% SDS-PAGE and transferred to a PVDF membrane (Millipore, Bedford, MA, USA). The membranes were blocked with 5% nonfat milk for 1 h at room temperature and then incubated at 4 °C overnight with primary antibodies. All primary antibodies were from Cell Signaling Technology (Danvers, MA, USA) and diluted 1:1000, as directed. The next day, the membranes were rinsed with TBS and incubated with secondary antibodies for 1 h. The bands were visualized using the Uvitec Cambridge gel imaging system (UVITEC Ltd, Cambridge, UK). The relative density of each protein band was determined using ImageJ software (Rawak Software, Inc., Stuttgart, Germany).

### 4.9. Antitumor Effect and Lung Metastasis in Orthotopic Breast Cancer Model

4T1-Luc cells were resuspended in 1:1 serum-free DMEM medium and Matrigel basement membrane matrix, and injected into the left 4th mammary fat pad of 6-week-old female BALB/c mice. Then, the tumor-bearing mice were randomly divided into five groups (*n* = 10 per group): Control (physiological saline, ig); 50 mg/kg Sorafenib, (in PEG200 with 10% (*v*/*v*) DMSO, ig); and 10 mg/kg, 20 mg/kg and 40 mg/kg compound **1** (in PEG200 with 10% (*v*/*v*) ethanol, ig). Each group was administered every other day during the experiment. The tumor volume and body weight were monitored and recorded. The tumors and main organs were excised from the sacrificed mice at the end of the experiment for hematoxylin and eosin (H&E) staining. To evaluate the pulmonary metastasis of 4T1 tumors, lungs were also isolated and imagined through bioluminescence imaging (BLI) and soaked in Bouin’s solution for photographs. In addition, angiogenesis of tumor tissues was evaluated by immunohistochemistry (IHC) analysis, and the microvessel density (MVD) was assessed [53].

### 4.10. Molecular Docking Simulation

The molecular docking simulation was examined as described previously [54]. The co-crystal structures of **1** with HSP90α (PDB ID: 4NH8) were prepared by Protein Preparation Wizard. The workflow provided in the MAESTRO r.2020-2 software (Schrödinger, LLC, New York City, NY, USA), using the OPLS3e force field.

### 4.11. Cellular Thermal Shift Assay (CETSA)

A total of 2 mL of MDA-MB-231 cell suspension (1.5 × 10^5^ /mL) was plated into each well of a 6-well plate. The next day, cells were treated with either **1** (15 μM), SNX-2112 (1 μM) or vehicle (DMSO, 0.1%) for 3 h. After being rinsed twice with cold PBS, cells were trypsinized, collected and centrifuged at 1000 r/min. The pellet was resuspended in PBS with protease inhibitors. The cell suspension was aliquot into PCR tubes with 50 μL per tube and heated at indicated temperatures for 3 min, followed by cooling at room temperature for another 3 min. Then, the cells underwent quick freezing and thawing 3 times in liquid nitrogen. The cell lysate was then centrifuged at 3000 r/min and the resulting supernatant was collected for the following immunoblotting assay.

### 4.12. Surface Plasmon Resonance (SPR) Analysis

The in vitro binding affinity assay between HSP90α and **1** was evaluated using a Biacore T200 surface plasmon resonance instrument (SPR, Biacore T200, GE healthcare, Chicago, IL, USA), as previously described [55]. A gradient concentration of **1** (0.25, 0.5, 1, 2, 4, 8, 16, 32 μM) was injected for analysis. Data were finally analyzed using the Biacore evaluation software (S200 version 1.0) by curve fitting using the 1:1 binding model, and the equilibrium dissociation constant (KD = kd/ka) was calculated.

### 4.13. Statistical Analysis

Statistical analyses were performed in GraphPad Prism 7.00 (GraphPad Software Inc., San Diego, CA, USA). The data are presented as the mean ± standard deviation (S.D.). Differences were considered statistically significant when *p* < 0.05 (* *p* < 0.05, ** *p* < 0.01 and *** *p* < 0.001).

## 5. Conclusions

To conclude, in the present study, we identified the pregnane alkaloid derivative **1** as an HSP90α inhibitor that could suppress breast cancer metastasis and angiogenesis by modulating the HIF-1α/VEGF/VEGFR2 pathway and the phosphorylation of downstream signaling molecules in vitro and in vivo. Therefore, compound **1** is a potential candidate for developing a novel antitumor agent for metastatic breast cancer.

## Figures and Tables

**Figure 1 molecules-27-07132-f001:**
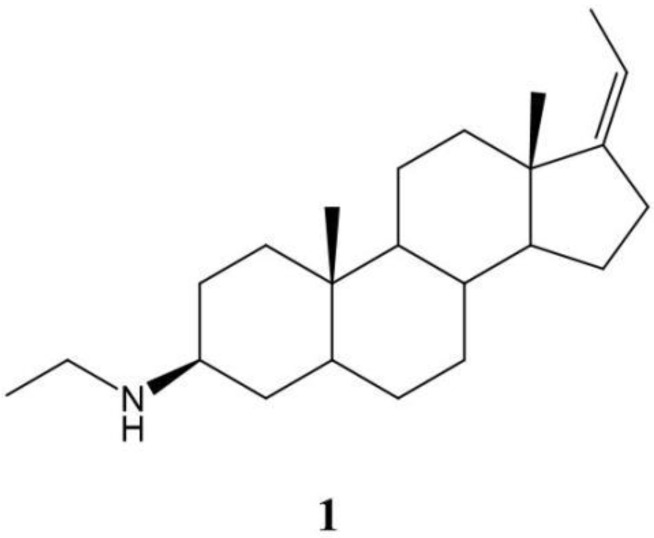
The chemical structure of compound **1**.

**Figure 2 molecules-27-07132-f002:**
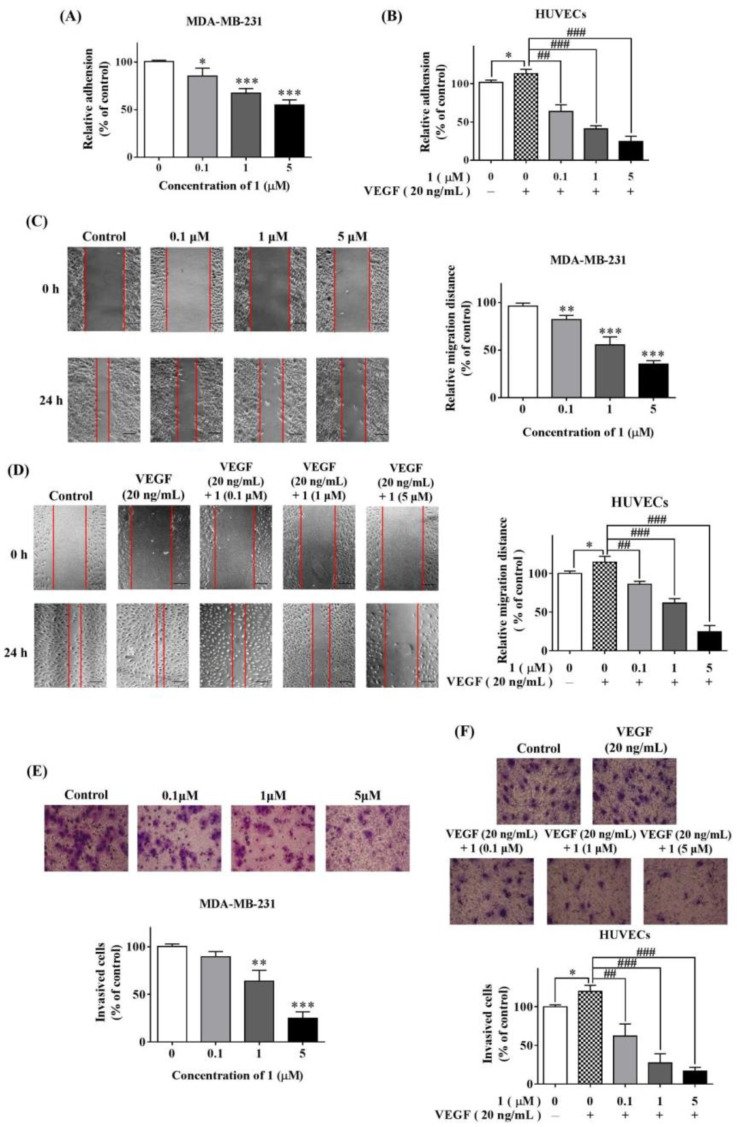
Compound **1** suppressed MDA-MB-231 cells and HVUECs adhesion, migration and invasion. The effect of **1** on MDA-MB-231 cells (**A**) and HUVECs (**B**) adhesion was determined by cell-adhesion assay. Wound-healing assay was performed to examine the role of **1** in suppressing MDA-MB-231 cells (**C**) and HUVECs (**D**) migration. **1** suppressed MDA-MB-231 cells (**E**) and HUVECs (**F**) invasion, which was assessed by transwell assay. (*n* = 3, means ± SEM are shown; * *p* < 0.05, ** *p* < 0.01, *** *p* < 0.001, compared with control (0.1% DMSO), ^##^ *p* < 0.01, ^###^ *p* < 0.001, compared with VEGF only).

**Figure 3 molecules-27-07132-f003:**
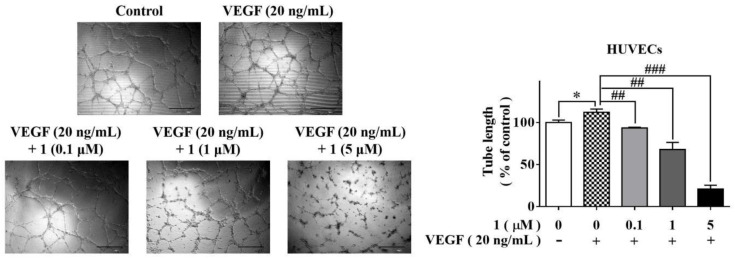
The effect of **1** on tube formation of HUVECs were examined by tube formation assay. (*n* = 3, means ± SEM are shown; * *p* < 0.05, compared with control, ^##^ *p* < 0.01, ^###^ *p* < 0.001, compared with VEGF only).

**Figure 4 molecules-27-07132-f004:**
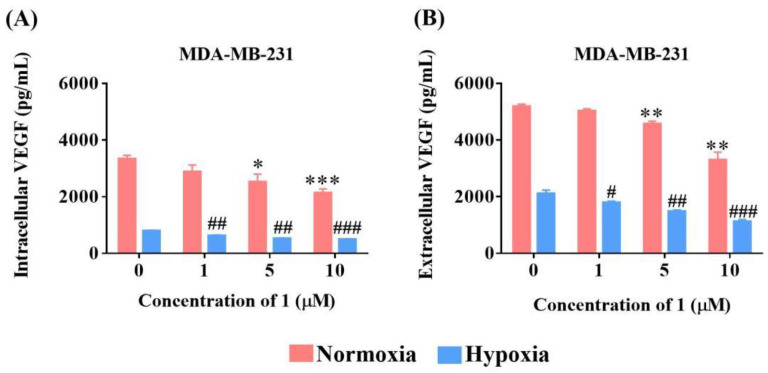
The effect of **1** on intracellular and extracellular VEGF expression in MDA-MB-231 cells. The inhibition of intracellular (**A**) and extracellular (**B**) VEGF expression level under hypoxia and normoxia were determined using human VEGF ELISA kit. (*n* = 3, means ± SEM are shown; * *p* < 0.05, ** *p* < 0.01, *** *p* < 0.001, compared with control under normoxia; ^#^
*p* < 0.05, ^##^ *p* < 0.01, ^###^
*p* < 0.001, compared with control under hypoxia).

**Figure 5 molecules-27-07132-f005:**
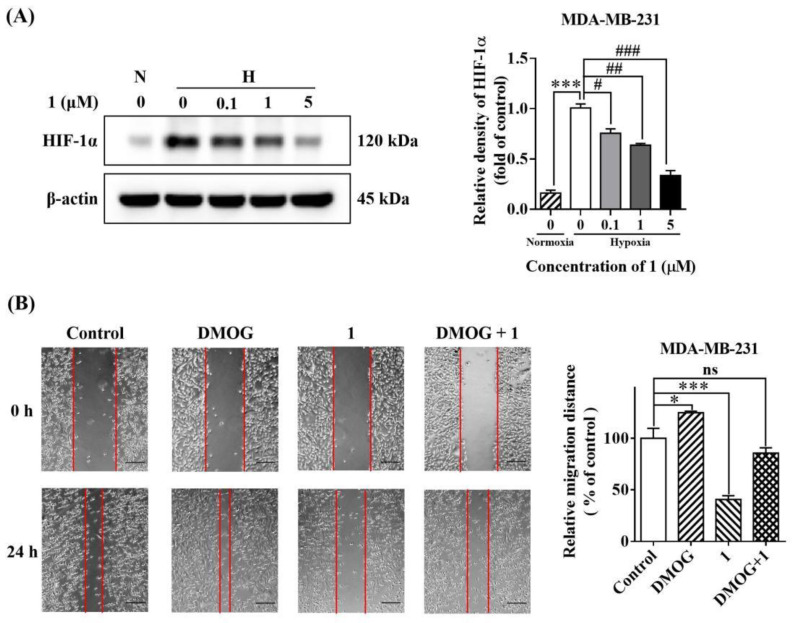
The effect of **1** on cell migration was correlated to HIF-1α expression. Compound **1** inhibited HIF-1α expression under hypoxia (**A**). (*n* = 3, means ± SEM are shown; *** *p* < 0.001, compared with control under normoxia; ^#^
*p* < 0.05, ^##^
*p* < 0.01, ^###^
*p* < 0.001, compared with control under hypoxia.) The HIF-1α agonist DMOG rescued MDA-MB-231 cell migration from the inhibition of **1** (**B**). (*n* = 3, means ± SEM are shown; * *p* < 0.05, *** *p* < 0.001, compared with control).

**Figure 6 molecules-27-07132-f006:**
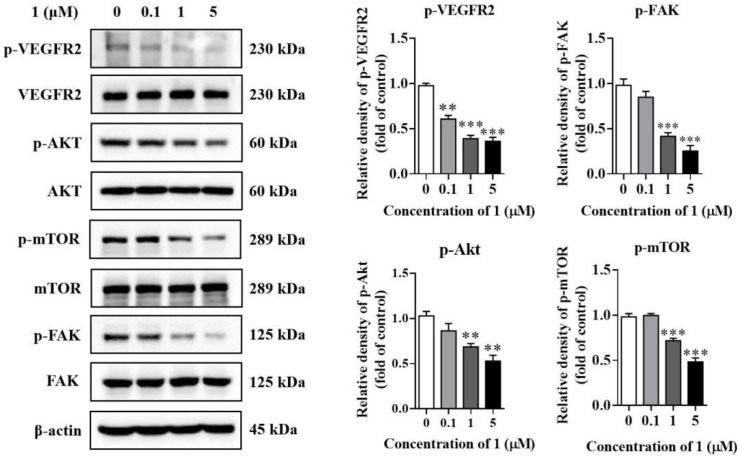
The effects of **1** on protein expression and phosphorylation levels of VEGFR2 and its downstream signaling pathways in breast cancer cells. The levels of p-VEGFR2, VEGFR2, p-Akt, Akt, p-mTOR, mTOR, p-FAK and FAK were determined by Western blots. (*n* = 3, means ± SEM are shown, ** *p* < 0.01, *** *p* < 0.001, compared with control).

**Figure 7 molecules-27-07132-f007:**
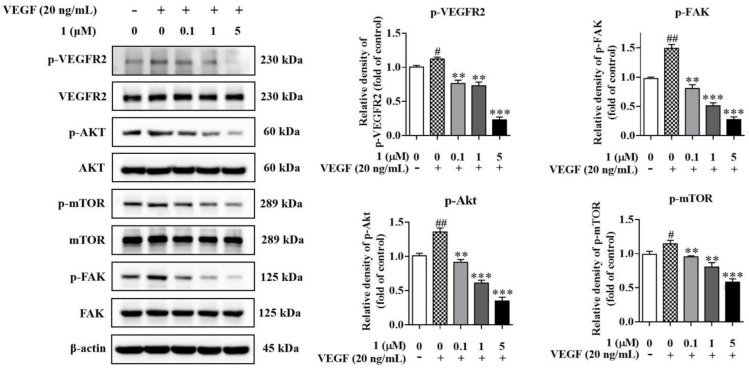
The effects of **1** on protein expression and phosphorylation levels of VEGFR2 and its downstream signaling pathways in HUVECs. The levels of p-VEGFR2, VEGFR2, p-Akt, Akt, p-mTOR, mTOR, p-FAK and FAK were determined by Western blots. (*n* = 3, means ± SEM are shown, ^#^ *p* < 0.05, ^##^ *p* < 0.01, compared with control; ** *p* < 0.01, *** *p* < 0.001, compared with VEGF only).

**Figure 8 molecules-27-07132-f008:**
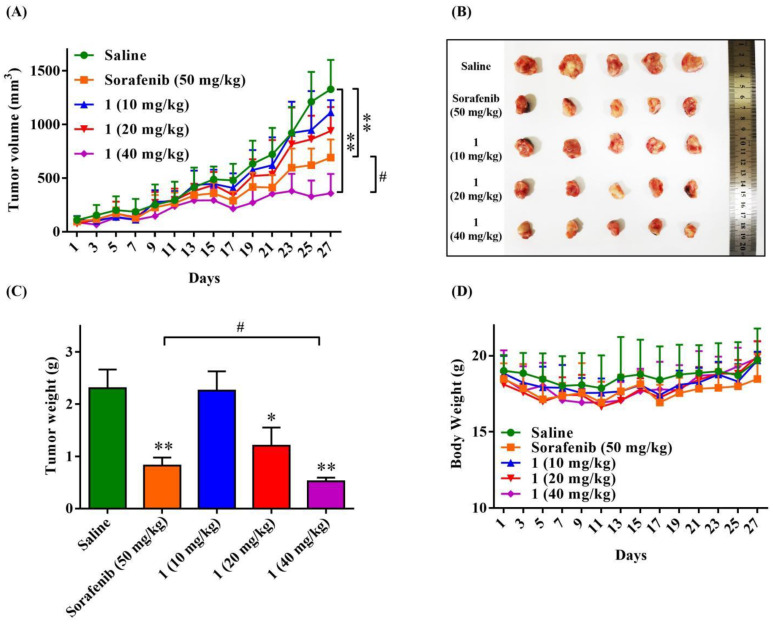
Compound **1** suppressed the tumor growth and metastasis in 4T1 mammary carcinoma model. Average tumor growth curves of each treatment group in 4T1 tumor-bearing mice (**A**). Tumor image (**B**) and tumor weight (**C**) of tumors excised from 4T1 tumor-bearing mice after treatment for 29 days. (*n* = 10, means ± SEM are shown, * *p* < 0.05, ** *p* < 0.01, compared with Saline treatment group; ^#^ *p* < 0.05, compared with Sorafenib treatment group.) Curves showing the body weight change of mice under various treatments in 4T1 tumor-bearing mice (**D**).

**Figure 9 molecules-27-07132-f009:**
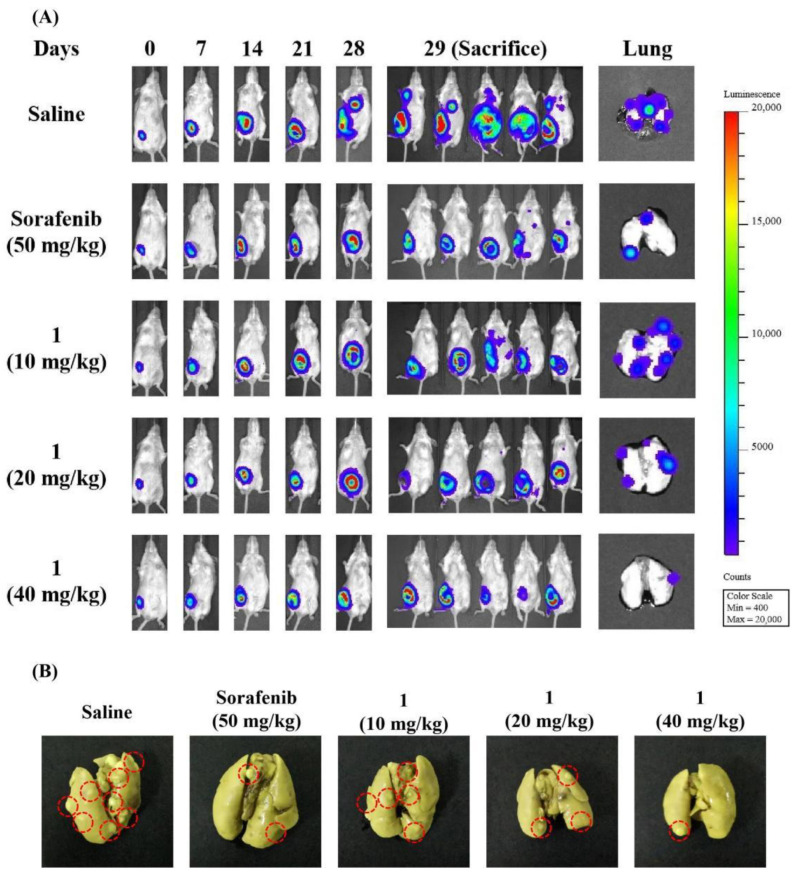
Compound **1** suppressed the tumor metastasis and angiogenesis in 4T1 mammary carcinoma model. Representative bioluminescent images showing primary (mammary fat pad) and metastatic (lung) breast tumors at different time points (0, 7, 14, 21, 28 and 29 days), and excised lungs at the end of the experiment (**A**). Bouin’s solution-filled images of lung (**B**) and the number of lung metastatic nodules in different treatment groups (**C**). IHC staining of CD31 in tumor tissues (**D**) with the microvessel density (MVD) (**E**). H&E staining of lung tissues in each group (**F**). (*n* = 5, means ± SEM are shown, * *p* < 0.05, ** *p* < 0.01, *** *p* < 0.001, compared with Saline treatment group; ^#^
*p* < 0.05, compared with Sorafenib treatment group.) Scale bar = 200 μm.

**Figure 10 molecules-27-07132-f010:**
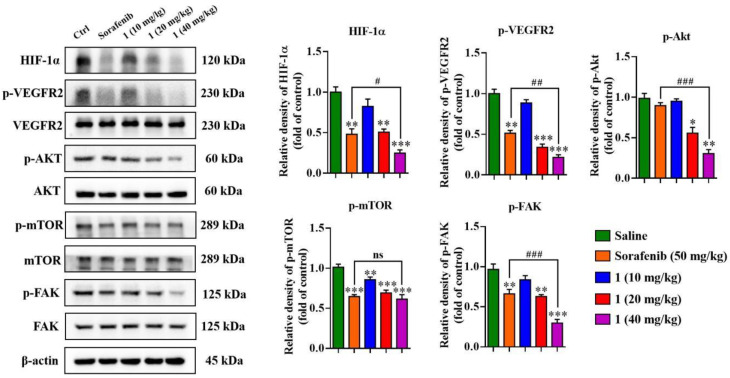
The effects of **1** on protein expression and phosphorylation levels of HIF-1α, VEGFR2 and its downstream signaling pathways of tumor tissue in 4T1 mammary carcinoma model. The levels of HIF-1α, p-VEGFR2, VEGFR2, p-Akt, Akt, p-mTOR, mTOR, p-FAK and FAK were determined by Western blots. (*n* = 3, means ± SEM are shown, * *p* < 0.05, ** *p* < 0.01, *** *p* < 0.001, compared with Saline treatment group; ^#^ *p* < 0.05, ^##^ *p* < 0.01, ^###^ *p* < 0.001, compared with Sorafenib treatment group).

**Figure 11 molecules-27-07132-f011:**
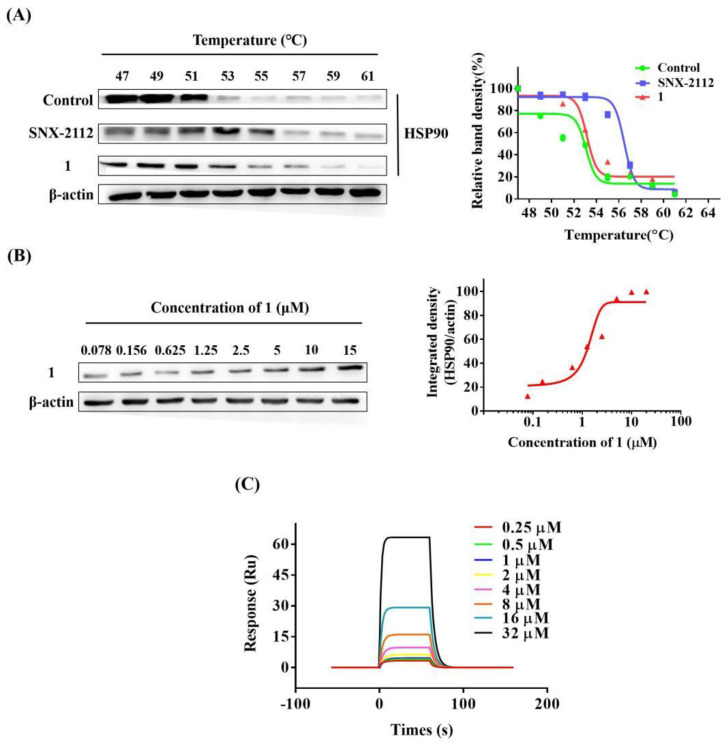
HSP90α was the direct target of **1**. Compound **1** bound directly to HSP90 in CESTA assay (**A**), and the binding was dose dependent (**B**). Real time binding affinity measurements of **1** (**C**) further demonstrated the direct target was HSP90α.

## Data Availability

The data that support the findings of the present study are available from the corresponding authors upon reasonable request.

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
