# Peer review of "Alkaloid Derivative (Z)-3β-Ethylamino-Pregn-17(20)-en Inhibits Triple-Negative Breast Cancer Metastasis and Angiogenesis by Targeting HSP90α"

_molecules, 2022, doi:10.3390/molecules27207132_

Round 1
Reviewer 1 Report
Xin-Yao Liu and colleagues presented a good piece of data demonstrating the role of pregnane alkaloid 1 in downregulating HIF-1α and blocking the VEGF/VEGFR2 axis. Furthermore, they also discuss the downstream signalling pathways affected by the compound in triple-negative breast cancer cells. Further their study also indicating the direct interaction of compound 1 with HSP90α. However, the inclusion of the following mentioned suggestion can further improve the scientific depth of the manuscript.
1. The authors must strictly check for the English and grammatical mistakes. For example, In the title of their paper the word “Matastasis” has been spelled incorrectly and should be corrected. It should be corrected to “metastasis”.
2. Author should improve on the introduction part of breast cancer and discussion part. Some examples are here PMID: 20564126; Cells. 2020 Jun 21;9(6):1511; Authors can discuss other HSPs in the discussion to make it more scientific.
3. Only the first letter of the first word of the paper title should be written in capital. The other words of the title should be corrected.
4. The authors have carried out their research on triple-negative breast cancer (TNBC) Cells. 2020 Jun 21;9(6):1511. I suggest that the same can be included/mentioned in the paper title instead of just “breast cancer”.
5. On Page 2 and Figure 1, please define the legend/caption. “The chemical structure of 1” sounds incomplete. The authors can replace “1” either with its complete nomenclature or with “Compound 1” / “Derivative 1”.
6. On Page 2, in Line 65, the authors have mentioned that the IC50 of Compound 1 is 0.17µM. It would be great if the corresponding/supporting/graphical data could be added to the research paper.
7. On Page 4, in Figure C, the authors should clarify whether the “Control” used for the experiment consists of only media or DMSO too.
8. Migration and invasion assays have been tested for the effect of 0.1 µM, 1 µM, and 5 µM of Compound 1 (Figures C, D, and Figures E, F). Whereas, the IC50 mentioned earlier was 0.17µM. The authors may kindly define the rationale for selecting this particular concentration range for experimental assays.
9. On Page 6, in Figures 4A and 4B, the name of the cell line (MDA-MB-231) experimented on should be mentioned in both the graphs as well as the figure caption/legend.
10. On Page 9, in Figure B, the sizes of tumors formed after treatment with 20mg/kg and 40mg/kg show only a little difference. Can the authors explain why is that so?
11. On Page 9, the authors should correct the grammar for lines 205 and 206.
“Notably, there were fewer lung nodules THAT existed…” Or
“Notably, fewer lung nodules existed…”
12. Point 8 holds the same for Figures 10 and 11A-11C on Pages 12 and 13, respectively.
13. 4T1 is a breast cancer cell line and is derived from a BALB/c strain (immunocompetent) mouse mammary gland. Were these experiments replicated in the SCID mice model too?
14. HUVECs are cells derived from the umbilical cord vein endothelium and are majorly/mostly used for in-vitro assessments. MCF 10A, a normal human breast epithelial cell line definitely would serve a better purpose for the documented study as the control. Why did the authors choose former in place of the latter?
Author Response
- The authors must strictly check for the English and grammatical mistakes. For example, In the title of their paper the word “Matastasis” has been spelled incorrectly and should be corrected. It should be corrected to “metastasis”.
Response: It was revised.
- Author should improve on the introduction part of breast cancer and discussion part. Some examples are here PMID: 20564126; Cells. 2020 Jun 21;9(6):1511; Authors can discuss other HSPs in the discussion to make it more scientific.
Response: It was revised. (As shown in manuscript marked by yellow color.)
- Only the first letter of the first word of the paper title should be written in capital. The other words of the title should be corrected.
Response: It was revised.
- The authors have carried out their research on triple-negative breast cancer (TNBC) Cells. 2020 Jun 21;9(6):1511. I suggest that the same can be included/mentioned in the paper title instead of just “breast cancer”.
Response: It was revised.
- On Page 2 and Figure 1, please define the legend/caption. “The chemical structure of 1” sounds incomplete. The authors can replace “1” either with its complete nomenclature or with “Compound 1” / “Derivative 1”.
Response: It was revised.
- On Page 2, in Line 65, the authors have mentioned that the IC50of Compound 1 is 0.17µM. It would be great if the corresponding/supporting/graphical data could be added to the research paper.
Response: According to the reviewer’s suggestion, the IC50 data of compound 1 was added to Supplementary Materials (As shown in Table S1).
- On Page 4, in Figure C, the authors should clarify whether the “Control” used for the experiment consists of only media or DMSO too.
Response: It was revised.
- Migration and invasion assays have been tested for the effect of 0.1 µM, 1 µM, and 5 µM of Compound 1 (Figures C, D, and Figures E, F). Whereas, the IC50 mentioned earlier was 0.17µM. The authors may kindly define the rationale for selecting this particular concentration range for experimental assays.
Response: Firstly, the screen method was different with above mentioned migration and invasion assay. Chemotaxis chamber was used to the preliminary screening in previous research. Secondly, the cell line of MDA-MB-231 was obtained from different companies. The previous study was carried out in 2015, and the present study was finished in March, 2022.
As well known, based on the different souce (or market cell lines), the results of phenotypic screening may appear difference.
- On Page 6, in Figures 4A and 4B, the name of the cell line (MDA-MB-231) experimented on should be mentioned in both the graphs as well as the figure caption/legend.
Response: It was revised.
- On Page 9, in Figure B, the sizes of tumors formed after treatment with 20mg/kg and 40mg/kg show only a little difference. Can the authors explain why is that so?
Response: The two-dimensional image was taken from the top view, which was influenced by placement angels . So it might cause the visual difference.
- On Page 9, the authors should correct the grammar for lines 205 and 206.
“Notably, there were fewer lung nodules THAT existed…” Or
“Notably, fewer lung nodules existed…”
Response: It was revised.
- Point 8 holds the same for Figures 10 and 11A-11C on Pages 12 and 13, respectively.
Response: Figure 10 was obtained from in vivo experiments, and the mice was administrated orally. Due to various reasons (bioavailability, drug metabolism, etc. ), effective dosages are difference. The dosages in this research were determined by pre-experiments. Figure 11C indicated the affinity of compound 1 with HSP90α. The results demonstrated that HSP90α was one of the direct targets of compound 1. However, the KD value of compound 1 with HSP90α was not in corresponding to phenotypic screening (migration, invasion, etc.). It deduced that compound 1 has an another target which also regulated HIF-1α.
- 4T1 is a breast cancer cell line and is derived from a BALB/c strain (immunocompetent) mouse mammary gland. Were these experiments replicated in the SCID mice model too?
Response: SCID mice is an immunodeficient model and is very suitable for evaluating the tumor metastasis. However, except for anti-metastasis effect, compound 1 also has cytotoxicity and anti-angiogenesis effect on breast cancer. In order to evaluate the anti proliferative, anti-metastasis and anti-angiogenesis effects, we chose the orthotopic 4T1 mammary carcinoma models for the in vivo experiments.
The continued research will be conducted by using SCID mice in our lab.
- HUVECs are cells derived from the umbilical cord vein endotheliumand are majorly/mostly used for in-vitro assessments. MCF 10A, a normal human breast epithelial cell line definitely would serve a better purpose for the documented study as the control. Why did the authors choose former in place of the latter?
Response: HUVECs were majorly/mostly used for in-vitro experiments on angiogenssis (especially for tube formation assay). HUVECs also were the most important cells in neovascularization process. Compound 1 is a derivative from natural source, we hope that compound 1 has comprehensive anti-angiogenesis effect. Thus, HUVECs was chosen for the first evaluation.
Reviewer 2 Report
Reviewers’ comments for the Manuscript ID: Molecules-1964436
In the current manuscript’s authors reported the in vitro and in vivo effect of “(Z)-3β-ethylamino-pregn-17(20)-en (1) synthetic analogue of pregnane alkaloid”. On cancer metastasis, angiogenesis, and underlying mechanism. This is well designed and written paper and suitable for publication in “Molecules after addressing below few general comments.
1) Introduction part is very brief, needs to expand it.
2) It is not clear why authors choose compound 1 (IC50 = 0.17 uM) in this study instead of most potent compound 20f (IC50 = 0.03uM) according to their previous publication reference [16].
3) In figure 2, E (control) image seems to be copied from previous publication (16) image figure-2, authors need to correct it, each experiment should contain independent controls.
Author Response
1) Introduction part is very brief, needs to expand it.
Response: It was revised (As shown in manuscript marked by yellow color.)
2) It is not clear why authors choose compound 1(IC50 = 0.17 uM) in this study instead of most potent compound 20f (IC50 = 0.03uM) according to their previous publication reference [16].
Response: Before the formal research begin, a pre-experiment of compounds 1 and 20f was conducted on tumor bearing mice. Unfortunately, compound 1 revealed more effective than that of 20f in vivo (may attribute to bioavailability and so on). Thus, compound 1 was chosen for further research.
3) In figure 2, E (control) image seems to be copied from previous publication (16) image figure-2, authors need to correct it, each experiment should contain independent controls.
Response: It was revised.